# Postmortem-Derived Exosomal MicroRNA 486-5p as Potential Biomarkers for Ischemic Heart Disease Diagnosis

**DOI:** 10.3390/ijms25179619

**Published:** 2024-09-05

**Authors:** So-Yeon Kim, Sookyoung Lee, Jong-Tae Park, Su-Jin Lee, Hyung-Seok Kim

**Affiliations:** 1Department of Forensic Medicine, Chonnam National University Medical School, Gwangju 61469, Republic of Korea; nfs_75@daum.net (S.-Y.K.); jtpark@jnu.ac.kr (J.-T.P.); 2Department of Forensic Medicine, National Forensic Service, 10, Ipchun-ro, Wonju-si 61469, Republic of Korea; heart@korea.kr

**Keywords:** exosomes, microRNAs, postmortem diagnostics, acute myocardial infarction, biomarkers

## Abstract

Exosomes are nanovesicles 30–150 nm in diameter released extracellularly. Those isolated from human body fluids reflect the characteristics of their cells or tissues of origin. Exosomes carry extensive biological information from their parent cells and have significant potential as biomarkers for disease diagnosis and prognosis. However, there are limited studies utilizing exosomes in postmortem diagnostics. In this study, we extended our initial research which identified the presence and established detection methodologies for exosomes in postmortem fluids. We analyzed exosomal miRNA extracted from plasma and pericardial fluid samples of a control group (n = 13) and subjects with acute myocardial infarction (AMI; n = 24). We employed next-generation sequencing (NGS) to investigate whether this miRNA could serve as biomarkers for coronary atherosclerosis leading to acute myocardial infarction. Our analysis revealed 29 miRNAs that were differentially expressed in the AMI group compared to the control group. Among these, five miRNAs exhibited more than a twofold increase in expression across all samples from the AMI group. Specifically, miR-486-5p levels were significantly elevated in patients with high-grade (type VI or above) atherosclerotic plaques, as per the American Heart Association criteria, highlighting its potential as a predictive biomarker for coronary atherosclerosis progression. Our results indicate that postmortem-derived exosomal microRNAs can serve as potential biomarkers for various human diseases, including cardiovascular disorders. This finding has profound implications for forensic diagnostics, a field critically lacking diagnostic markers.

## 1. Introduction

Myocardial infarction, resulting from atherosclerotic ischemic heart disease, is likely the most common diagnosis in the majority of sudden-death cases subjected to clinical and medicolegal autopsies. This condition typically arises from a reduction in blood supply to the myocardium due to occlusion or narrowing of coronary arteries, often attributable to atherosclerosis, thrombus, or vascular spasm [1,2]. Specifically, ischemic heart disease, including acute myocardial infarction, was reported to account for the largest part among sudden cardiac death [3]. According to an autopsy-based study in Korea, 48.7% of natural deaths were associated with sudden cardiac death [3]. Clinical diagnosis facilitated by tools such as electrocardiograms, echocardiograms, coronary angiography, and cardiac biomarkers also allow for effective monitoring of disease status. Forensic diagnosis is challenging due to postmortem changes, making it impossible to use blood biomarkers commonly used in clinical settings. Therefore, the degree of coronary artery occlusion, the presence of thrombi, and ischemic changes or fibrosis of the myocardium are assessed during the autopsy. And the changes in myocardium, the extent of fibrosis and inflammatory cell infiltration are examined through histopathological analysis [4,5]. Even if all of the above methods are used, if the myocardial changes are not clear, diagnosing myocardial infarction may be difficult. This is why additional biomarkers are needed to increase the accuracy of forensic diagnosis. According to Kuninaka et al. and Hiyamizu et al., the expression of Nrf2 and HO-1 in postmortem cardiac tissue were increased in acute myocardial ischemia, which may be helpful in the diagnosis of acute myocardial infarction [6,7].

Exosomes are extracellular vesicles, approximately 30–150 nanometers in size, released from the plasma membrane via the endocytic pathway [8,9]. Exosome protect biomolecules such as proteins, DNA, and RNA from enzymatic degradation within their bilayer membrane and deliver these cargos to target cells [10,11]. They can be detected in all body fluids, including cerebrospinal fluid, saliva, breast milk, blood, and urine. Exosomes extracted from each body fluid reflect the characteristics of the corresponding organ [11,12].

Pericardial fluid exosomes from acute myocardial infarction (AMI) patients contain biomarkers important for the development and progression of ischemic heart disease, such as let-7b-5p, miR-21-5p, and miR-22-3p [13]. The number of exosomes released increases under conditions such as hypoxia and inflammation [14]. These findings suggest the potential application of exosomes in distinguishing ischemic injury caused by disease from ischemia of the whole body during the process of death. In our previous study, we confirmed the successful isolation of exosomes from postmortem body fluids. Postmortem exosomes exhibited no significant morphological differences compared to antemortem exosomes and were well preserved even under simulated degradation conditions [15].

MicroRNA (miRNA) is a non-coding RNA approximately 22 nucleotides in length [16]. miRNA functions in post-transcriptional processes by guiding Argonaute proteins to the 3′UTR region of target mRNA, suppressing gene translation, and regulating expression [17]. Since the discovery of miRNA in exosomes by Valadi et al. [18], research on the functions of circulating miRNA and its potential as a biomarker for disease diagnosis has been actively pursued [19,20,21]. In forensic medicine, miRNA is currently used to identify body fluids with tissue specificity [22,23,24,25,26], but research aiding in diagnosing the cause of death is not yet extensive [27].

The purpose of this study is to overcome the limitations of existing biomarkers, which rapidly deteriorate and decompose after death, and to discover novel biomarkers for diagnosing acute myocardial infarction based on exosomes.

## 2. Results

### 2.1. Quality of Exosomal RNA Isolated from Postmortem Plasma and Pericardial Fluid

The quality of exosomal RNA was assessed by the absorbance spectrum. Notable differences between sample types were observed in peak patterns. Exosomes derived from plasma typically exhibited a sharp peak at 25 nt (Figure 1A,B), whereas those extracted from pericardial fluid showed a broader range of smaller peaks following a prominent peak (Figure 1C,D). The pattern of peaks differed only between plasma and pericardial fluid, and no differences were observed between the control and MI groups within each sample type. 

### 2.2. Analysis of Small RNA Profiling of Postmortem Exosomes Using Next-Generation Sequencing

Significant expression levels were defined as those with an increase or decrease in more than twofold in the MI group compared to the control group with a *p*-value of less than 0.05. There were 29 miRNAs that showed significant expression levels in the MI group compared to the control group. miR-486-5p had the highest average read count, ranging from a minimum of 16 to a maximum of 3627 (Table 1). Read counts for most miRNAs were measured at low levels.

In plasma-derived exosomes, five miRNAs were significantly increased more than twofold in the MI group compared to the control group. miR-3591-5p exhibited the highest increase at 11.929-fold, while miR-3184-3p showed the smallest increase at 2.516-fold (Figure 2A). In pericardial-fluid-derived exosomes, two miRNAs demonstrated more than a twofold increase in the MI group compared to the control group: miR-4443 increased by 5.503-fold, and miR-625-3p increased by 5.053-fold. Additionally, 12 miRNAs in pericardial-fluid-derived exosomes showed a significant decrease of more than twofold (Table 2).

### 2.3. Validation of Significantly Increased miRNA

For more accurate statistical results, qRT-PCR was conducted with additional 10 control samples and 19 MI samples. However, it was challenging to identify a statistically significant increasing or decreasing trend in gene expression results (Figure 2B). Additionally, qRT-PCR was performed to determine whether there was a difference in the expression of miR-486-5p in whole-blood samples used in the previous NGS experiment. The result showed no significant difference in expression between each group in whole blood (Figure 3).

### 2.4. miR-486-5p Expression in Advanced Atherosclerotic Lesions

According to the AHA classification criteria, in atherosclerotic lesions of the left anterior descending artery with a grade higher than lesion type V, we observed complex plaques with intraplaque hemorrhage or calcification (Figure 4A). In the plasma-derived exosomes of the high-grade atherosclerosis group, the expression levels of cardiac blood-derived exosomal miR-486-5p were statistically significantly upregulated compared to the low-grade atherosclerosis group (*p* = 0.0216) (Figure 4B).

## 3. Discussion

Elucidating the mechanisms that occur in our bodies after death and harnessing rapidly degrading biological information remains a challenging task for forensic researchers. The ultimate goal of this study was to identify exosome-based biomarkers for acute myocardial infarction (AMI) that are applicable even in postmortem conditions.

The RNA integrity number (RIN) values of autopsy samples were measured to be low, and typically, the 28S/18S ratio was not considered in evaluating the quality of exosomal RNA. Unlike conventional RNA samples, exosomes do not contain RNA as large as ribosomal RNA. Instead, the quality of exosomal RNA was assessed based on migration patterns and peaks. When comparing the RNA patterns between plasma-derived and pericardial-fluid-derived exosomes, it was observed that pericardial-fluid-derived exosomes exhibited a greater diversity of small RNA sizes [28]. Pericardial fluid is the serous fluid in the pericardial sac that surrounds the heart. This fluid has different properties (exudative or transudative) and amounts depending on the disease state. 

Because the mechanism of pericardial fluid production is different from plasma, it can contain larger exosomes, and its amount appeared to be increased in the MI group. Unfortunately, an appropriate endogenous control for exosomal miRNA analysis remains elusive. Although U6 or miR-16 is utilized for qRT-PCR normalization in certain studies, substantial controversy surrounds their stability and cellular localization [29]. Thus, in this study, we compared the absolute expression levels between the control group and the MI group.

The verification of miRNA candidate expression selected through NGS using qRT-PCR revealed challenges in identifying a consistent trend of increase or decrease that aligns with the NGS results. Particularly, research on the expression of exosomal microRNA in postmortem serum for diagnosing myocardial infarction has only recently been reported, with a very limited sample size (n = 3 for control and MI cases, respectively) [27]. In comparison, our study performed NGS and qRT-PCR on a significantly larger number of cases (13 control cases and 24 MI cases).

Currently, research on circulating miR-486-5p in AMI and its expression in the myocardium remains insufficient. Specifically, Wei et al. reported miR-486-3p as a stable marker distinguishing patients with three-month ST-elevated acute myocardial infarction (AMI) from those with stable ischemic heart disease [30]. Additionally, Niculescu et al. suggested that miR-486 could be utilized as an additional tool to differentiate vulnerable coronary artery disease patients [31]. In line with these studies, we observed a significant increasing trend of exosomal miR-486-5p in the high-grade atherosclerosis group based on the classification of the American Heart Association (*p* = 0.0216), while no distinct difference was found between groups in whole blood. This suggests that miR-486-5p, as an exosome-specific biomarker predicting high-risk factors such as complicated plaques, could be employed for postmortem diagnosis of AMI.

Despite the valuable insights gained from this study, it has limitations. Future studies will identify target genes regulated by miR-486-5p, which have not been clearly elucidated to date. Additionally, we seek to investigate the physiological functions of these target genes, aiming to comprehend the mechanistic actions of miR-486-5p in acute myocardial infarction. Biological experiments using autopsy samples present various challenges. However, sustained interest and research in biomarkers applicable even after death are anticipated to contribute to identifying the cause of death in forensic science. Furthermore, this research may prove beneficial for diagnosing diseases in living individuals.

In conclusion, our NGS analysis revealed the presence of miRNA indicating differences between the myocardial infarction and control groups in both plasma-derived and pericardial-fluid-derived exosomes collected during autopsy. Although the results of NGS were not replicated in qRT-PCR, we observed a significant increase in miR-486-5p in plasma-derived exosomes of the high-grade atherosclerosis group classified according to AHA criteria. Moreover, there was no significant intergroup difference in miR-486-5p in whole blood. This suggests the potential utility of miR-486-5p as a specific exosomal marker for diagnosing postmortem acute myocardial infarction.

## 4. Materials and Methods

### 4.1. Sample Collection and Preparation

Plasma derived from cardiac blood and pericardial fluid used in the experiment was collected through autopsies conducted at Chonnam National University from 2020 to 2022. The collection and use of these samples for this study were approved by the National Forensic Service Institutional Review Boards (permission #: 906-220421-BR-008-02). The samples were classified into control and myocardial infarction (MI) groups. The control group consisted of autopsy cases in which ischemic heart disease was not the cause of death and there were no vascular risk factors (diabetes, hypertension, and smoking). The MI group included autopsy cases of acute myocardial infarction based on autopsy findings, where significant coronary artery atherosclerosis and myocardial hypertrophy were observed. Additionally, these MI cases were finally diagnosed when histopathological examination revealed either more than 70% coronary artery stenosis or the presence of a thrombus, along with marked interstitial fibrosis in the myocardium. The percentage of luminal stenosis was calculated by comparing the remaining lumen area with the original lumen area (Table 3).

Cardiac blood was transferred to EDTA vacuum tubes (Greiner Bio-One, Kremsm ünster, Austria), while pericardial fluid was transferred to 15 mL conical tubes (SPL Life Science CO., Ltd., Pocheon, Republic of Korea). On the day of collection, platelet-free plasma was obtained by repeated centrifugation at 1550× *g* for 30 min and 3200× *g* for 30 min to remove cells [32]. The plasma and pericardial fluid were filtered through a 0.45 μm syringe filter (Corning Inc., Corning, NY, USA). The samples not used immediately in the experiment were stored at −80 °C. 

### 4.2. Information of Samples Used for Experiments

For next-generation sequencing (NGS), plasma and pericardial fluid samples from the control group (n = 3) and the MI group (n = 5) were used. Additionally, samples from a larger group of controls (n = 10) and MI patients (n = 19) were used for qRT-PCR analysis.

The autopsied left anterior descending artery (LAD) was classified into low (AHA class I to V)- and high (AHA class VI to VIII)-risk groups according to the Modified American Heart Association classification [33,34,35]. Table 1 shows the gender, age, postmortem interval (PMI), body mass index (BMI), risk factors related to cardiovascular diseases, heart weight, the percentage of luminal stenosis, and types of atherosclerotic lesions for each case. 

### 4.3. Exosomal RNA Isolation and Quality Control Analysis

Exosomal RNA was extracted from 250 μL of plasma and pericardial fluid using the exoRNeasy Midi Kit (Qiagen Inc., Hilden, Germany) according to the manufacturer’s instructions. Exosomal RNA was quantified using the NanoDrop 2000 Spectrophotometer system (Thermo Fisher Scientific, Waltham, MA, USA). The yield and distribution of small RNA were assessed using the Agilent 2100 Bioanalyzer with the RNA 6000 Pico Chip (Agilent Technologies, Amstelveen, The Netherlands) according to the manufacturer’s instructions.

### 4.4. Next-Generation Sequencing (NGS) Procedure

Library construction was performed using the NEBNext Multiplex Small RNA Library Prep Kit (New England BioLabs Inc., Ipswich, MA, USA) according to the manufacturer’s instructions. Briefly, total RNA from each sample was used for adaptor ligation, followed by cDNA synthesis using reverse transcriptase with adaptor-specific primers. Polymerase chain reactions (PCRs) were conducted for library amplification, and the libraries underwent cleanup using the QIAquick PCR Purification Kit (Qiagen Inc., Germany) and polyacrylamide gel electrophoresis (PAGE). The yield and size distribution of the small RNA libraries were assessed using the Agilent 2100 Bioanalyzer instrument (Agilent Technologies Inc., Santa Clara, CA, USA). High-throughput sequences were generated by the NextSeq500 system using single-end 75 sequencing (Illumina, San Diego, CA, USA). Sequence reads were mapped using the bowtie2 software v2.3.5.1tool to obtain the BAM file. The mature miRNA sequence served as a reference for mapping, and read counts mapped to the mature miRNA sequence were extracted from the alignment file using bedtools v2.25.0 and Bioconductor, the latter of which utilizes the R statistical programming language. The CPM + TMM normalization method was applied for comparison between samples. The NGS experiment was conducted by E-biogen Inc. (Seoul, Republic of Korea).

### 4.5. Validation Using Quantitative RT-PCR

miRNA was reverse-transcribed using a TaqMan™ Advanced miRNA cDNA Synthesis Kit (Thermo Fisher Scientific, Waltham, MA, USA) with a preamplification step according to the manufacturer’s recommendations. A total of 20 ng of total RNA was used as input for cDNA synthesis. Real-time PCR was performed with TaqMan™ Fast Advanced Master Mix (Thermo Fisher Scientific, Waltham, MA, USA) and TaqMan™ Advanced miRNA Assay (Thermo Fisher Scientific, Waltham, MA, USA) using a QuantStudio 3 Real-Time PCR system with a 96-well block module (Thermo Fisher Scientific, Waltham, MA, USA). The sequences for the qRT-PCR were as follows: miR-423-5p (5′-UGAGGGGCAGAGAGCGAGACUUU-3′), miR-486-5p (5′-UCCUGUACUGAGCUGCCCCGAG-3′), miR-625-3p (5′-GACUAUAGAACUUUCCCCCUCA-3′), miR-3591-5p (5′-UUUAGUGUGAUAAUGGCGUUUGA-3′), miR-4443 (5′-UUGGAGGCGUGGGUUUU-3′). All reactions were performed in triplicate for each sample.

### 4.6. Histopathology for Postmortem Human Coronary Artery

RNA extraction was performed on 13 control cases and 24 cases of myocardial infarction. Corresponding coronary artery samples were taken for each case, and a histopathological examination was conducted by preparing hematoxylin-and-eosin-stained slides. The coronary arteries were cannulated, washed with 0.1 mol/L PBS (pH 7.4), and perfused with 1L of freshly prepared 4% (wt/vol) paraformaldehyde in 0.1 mol/L sodium phosphate (pH 7.4) at 100 mm Hg. The left anterior descending coronary artery was dissected free from the surface of the heart, cut perpendicular to the long axis (from the proximal to distal segment) at 5 mm intervals, and then embedded in paraffin. Each section was stained with hematoxylin and eosin. In accordance with the definitions proposed by the Committee on Vascular Lesions of the Council on Arteriosclerosis, AHA, the atherosclerotic lesion type of each section was carefully classified by two investigators (K.H.S and L.S.J) simultaneously using a double-headed light microscope.

### 4.7. Statistical Analysis

Statistical analysis was performed with GraphPad Prism software v8.0.1. For comparison of expression values, the Mann–Whitney U test was used. *p*-values less than 0.05 were considered statistically significant. To calculate the fold change (FC), the negative ΔΔCT value is exponentiated by 2, as CT values are logarithmically related to the quantity of microRNA.

## Figures and Tables

**Figure 1 ijms-25-09619-f001:**
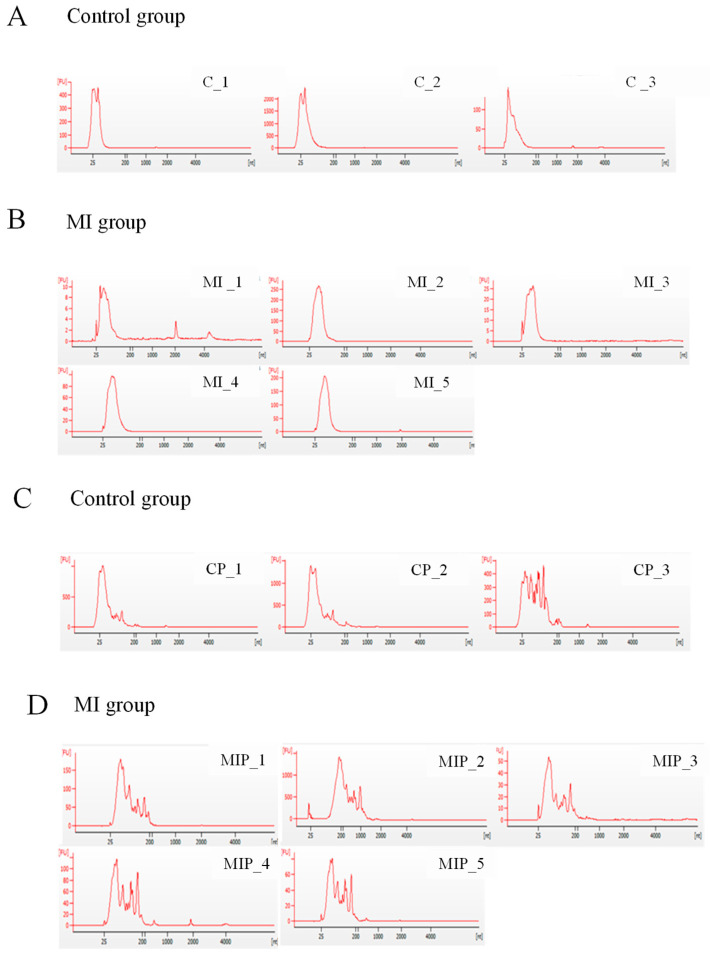
Quality control of postmortem exosomal RNA. The electropherograms display the peak distribution and fluorescence intensity (FU) of exosomal RNA extracted from cardiac blood and pericardial fluid. Panels (**A**,**B**) show the exosomal microRNA peaks from cardiac blood, while panels (**C**,**D**) illustrate those from pericardial fluid. The exosomal microRNA peaks are well observed in both sources; however, the peaks from cardiac blood are more distinct compared to those from pericardial fluid.

**Figure 2 ijms-25-09619-f002:**
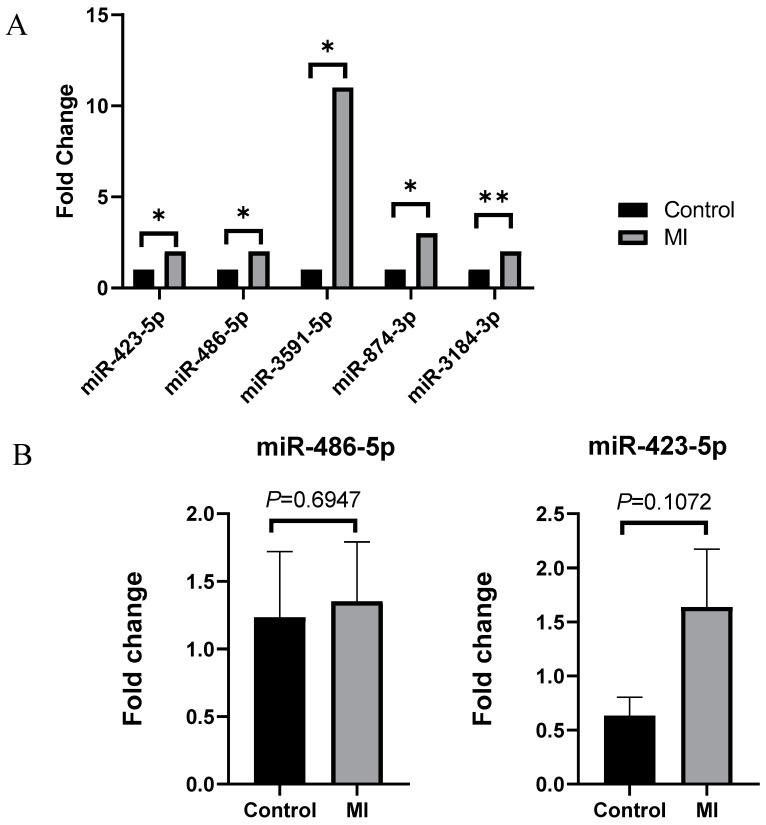
NGS and qRT-PCR results in plasma-derived exosomes from control and myocardial infarction groups. (**A**) Five miRNAs (miR-423-5p, miR-486-5p, miR-3591-5p, miR-874-3p, and miR-3184-3p) were significantly increased in plasma-derived exosomes from the myocardial infarction group compared to the control group. (**B**) The expression levels of miR-486-5p and miR-423-5p in additional plasma-derived exosome samples showed no statistically significant difference between the control and myocardial infarction groups. Values are presented as mean ± S.E.M. Statistical significance was determined using the Mann–Whitney U test. *: *p* < 0.05, **: *p* < 0.01.

**Figure 3 ijms-25-09619-f003:**
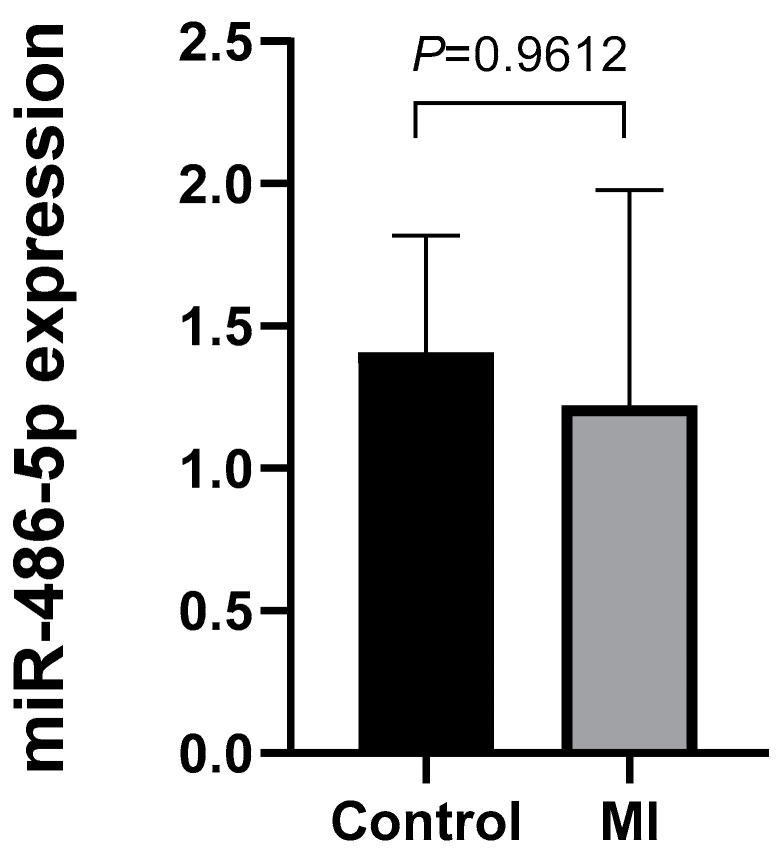
miR-486-5p expression in whole blood of control and myocardial infarction groups. The expression of miR-486-5p in whole blood did not show a significant difference between the control and myocardial infarction groups. Values are presented as mean ± S.E.M. Statistical significance was determined using the Mann–Whitney U test.

**Figure 4 ijms-25-09619-f004:**
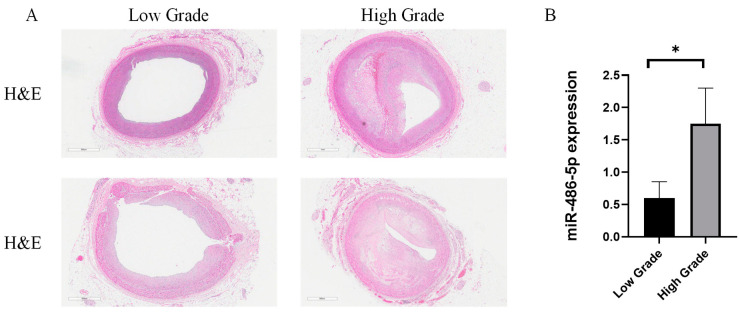
Histology of the LAD and expression of miR-486-5p in low- and high-grade atherosclerosis classified by the AHA. (**A**) Histological images show complex plaques in high-grade atherosclerosis involving hemorrhage or calcification (200× magnification). (**B**) The expression levels of cardiac blood-derived exosomal miR-486-5p were significantly higher in myocardial infarction (MI) cases with high-grade atherosclerosis compared to those with low-grade atherosclerosis, as classified by the American Heart Association (AHA). Values are presented as mean ± S.E.M. Statistical significance was determined using the Mann–Whitney U test (* *p* = 0.0216). AHA, American Heart Association.

**Table 1 ijms-25-09619-t001:** Differentially expressed miRNA of plasma-derived exosomes in MI group compared to control.

	miRNA ID	Control (log2)	Myocardial Infarction (log2)	Fold Change (H/CH)	*p*-Value
Up	hsa-miR-3591-5p	2.374	5.950	11.929	0.027
	hsa-miR-874-3p	3.845	5.575	3.319	0.024
	hsa-miR-423-5p	10.870	12.405	2.897	0.012
	hsa-miR-486-5p	13.363	14.698	2.522	0.045
	hsa-miR-3184-3p	10.918	12.249	2.516	0.009
Down	hsa-miR-4755-5p	5.183	0.000	0.028	0.017
	hsa-miR-554	4.762	0.000	0.037	0.018
	hsa-miR-4632-3p	5.916	2.006	0.067	0.030
	hsa-miR-6729-3p	6.337	2.540	0.072	0.033
	hsa-miR-1296-5p	4.762	1.363	0.095	0.026
	hsa-miR-7107-5p	5.586	2.483	0.116	0.005
	hsa-miR-3687	6.120	3.129	0.126	0.045
	hsa-miR-411-3p	4.969	2.051	0.132	0.008
	hsa-miR-1237-3p	4.762	2.006	0.148	0.043
	hsa-miR-6754-3p	4.969	2.967	0.250	0.050
	hsa-miR-28-3p	9.886	9.068	0.568	0.048

**Table 2 ijms-25-09619-t002:** Differentially expressed miRNA of pericardial-fluid-derived exosomes in MI group compared to control.

	miRNA ID	Control (log2)	Myocardial Infarction (log2)	Fold Change (H/CH)	*p*-Value
Up	hsa-miR-4443	8.389	10.849	5.503	0.009
	hsa-miR-625-3p	3.916	6.253	5.053	0.005
Down	hsa-miR-1295a	6.007	0.000	0.016	0.035
	hsa-miR-4494	6.007	0.000	0.016	0.035
	hsa-miR-212-5p	5.593	0.000	0.021	0.046
	hsa-miR-4781-3p	7.629	2.830	0.036	0.026
	hsa-miR-379-5p	7.035	2.354	0.039	0.029
	hsa-miR-5585-5p	6.658	2.354	0.051	0.029
	hsa-miR-214-3p	6.084	2.150	0.065	0.024
	hsa-miR-1287-5p	7.808	3.899	0.067	0.039
	hsa-miR-5095	7.676	5.002	0.157	0.050
	hsa-miR-4497	11.467	8.905	0.169	0.041
	hsa-miR-1285-3p	7.775	5.717	0.240	0.043
	hsa-miR-107	6.973	5.387	0.333	0.023

**Table 3 ijms-25-09619-t003:** Clinical characteristics of autopsy cases in this study.

Control Group (n = 13)											
No.	Age	Sex	BMI (kg/m)	PMI	Cause of Death	Heart Weight (g)	Risk Factors			Luminal Stenosis (%)	Lesion Type
							DM	HTN	Smoking		
1	52	F	22.2	26 h 10 m	Hanging	300	A ^+^	A	A	0	I
2	21	F	21.5	15 h 50 m	Sudden manhood death syndrome	264	A	A	A	0	I
3	47	F	19.4	75 h 30 m	Drug intoxication	276	A	A	A	56	III
4	70	F	23.4	42 h 10 m	Manual strangulation	336	A	A	A	0	I
5	65	M	26.7	65 h 40 m	Drug intoxication	460	A	A	A	83	VII
6	64	M	22.5	35 h 30 m	Drug intoxication	372	A	A	A	83	VI
7	56	F	21.4	39 h 30 m	Hanging	320	A	A	A	84	VI
8	58	M	16.5	53 h 50 m	Asphyxia	304	A	A	A	66	V
9	50	M	28.1	87 h 00 m	Subarachnoid hemorrhage	452	A	A	A	73	IV
10	74	F	19.9	65 h 30 m	Non-traumatic cerebral hemorrhage	402	A	A	A	62	IV
11	54	M	16.2	45 h 30 m	Drug intoxication	406	A	A	A	87	VII
12	25	M	27.4	86 h 00 m	Drug intoxication	344	A	A	A	0	I
13	59	F	22.1	25 h 10 m	Drug intoxication	372	A	A	A	0	Ⅰ
**MI Group (n = 24)**											
**No.**	**Age**	**Sex**	**BMI (kg/m)**	**PMI**	**Cause of Death**	**Heart** **Weight (g)**	**Risk Factors**			**Luminal Stenosis (%)**	**Lesion Type**
							**DM**	**HTN**	**Smoking**		
1	66	M	29.3	31 h 50 m	Ischemic heart disease including acute myocardial infarction	580	A	P	A	78	VII
2	52	M	31.7	39 h 20 m		680	A	A	A	88	V
3	54	M	26.2	21 h 40 m		412	A	A	A	100	VI *
4	55	M	26.7	24 h 20 m		426	A	A	A	88	V
5	49	M	24.8	48 h 55 m		388	A	A	A	86	V
6	59	M	22.4	37 h 50 m		378	A	A	A	95	V
7	35	M	21.4	48 h 15 m		370	A	A	A	76	VI
8	49	M	28.2	56 h 20 m		434	A	A	A	95	V
9	32	M	24.3	26 h 30 m		422	A	A	A	96	V
10	63	M	21	27 h 00 m		444	A	P	P	75	VII
11	55	M	24.8	42 h 05 m		460	A	A	A	88	VI
12	46	M	26.6	25 h 50 m		404	A	A	P	84	V
13	39	M	29.2	42 h 10 m		530	A	A	A	94	VI
14	76	M	15.3	39 h 00 m		362	A	A	A	100	V *
15	63	M	25.4	48 h 25 m		520	A	A	A	85	VI
16	55	M	25.1	97 h 05 m		408	A	A	A	82	VI
17	62	M	19.4	58 h 50 m		400	A	A	A	87	V
18	55	M	22.0	64 h 40 m		422	A	A	A	89	VI
19	52	M	27.4	26 h 30 m		764	A	A	A	80	V
20	57	M	25.5	67 h 50 m		420	A	A	A	75	VII
21	66	M	23.3	26 h 00 m		610	P	A	A	85	VII
22	59	F	18.9	97 h 45 m		446	A	A	A	91	VII
23	40	M	29.5	35 h 00 m		462	A	A	A	92	VII
24	44	M	22.5	29 h 00 m		452	A	A	A	75	VII

BMI, body mass index; PMI, postmortem interval; DM, diabetes mellitus; HTN, hypertension; h, hour; m, minute. * Luminal thrombus was identified (MI group: case numbers 3 and 14). + A: Absence, P: Presence.

## Data Availability

All relevant data are within the manuscript.

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
