# Peer review of "Postmortem-Derived Exosomal MicroRNA 486-5p as Potential Biomarkers for Ischemic Heart Disease Diagnosis"

_ijms, 2024, doi:10.3390/ijms25179619_

Round 1

Reviewer 1 Report

Comments and Suggestions for Authors

Interesting results from precious postmortem samples.

1. Introduction

1) Unlike clinical diagnosis, which relies on symptoms and various tests, autopsy diagnosis does not depend on the direct visualization of infarcted myocardial tissue. 

-> After "Unlike clinical diagnosis", there should be a difference between autopsy diagnosis and clinical diagnosis. But, 'not depend on the direct visualization of infarcted myocardial tissue' is not a difference, because, also in clinical diagnosis, visualization of infarcted myocardial tissue is very rare.

2) Exosomes protect cargos such as proteins, DNA, and RNA from enzymatic degradation within their bilayer membrane and deliver them to target cells.

-> Is this sentence correct? Do you mean proteins, DNA, and RNA are cargos protected by exocomes? Or Exosomes are protection cargos?

2. Material and methods

1) Please describe the criteria to diagnose the MI deaths. Although it was autopsy diagnosis by medical examiners, the description is required as inclusion criteria for this study.

2) In the description about Table 1, Table 1 should contain information on sex, age, PMI, BMI, risk factors related to cardiovascular diseases, heart weight, and type of atherosclerotic lesion. But, PMI, risk factors related to cardiovascular disease, and heart weight are missing. Please add them to the Table 1.

3) AHA classification represents maturity and advance of coronary atherosclerosis. It is expected to be correlated with the degree of occlusion, but not the same. How about include the percent of occlusion in your description of coronary atherosclerosis? The criteria for this percent occlusion can be found in a Book by Buja and Butany, Cardiovascular pathology 5th edition, pp8-10.

4) If there was thrombus or visible infarct in the ventricular wall in any MI case, it can be a good positive control. Wasn't there?

Author Response

[Reviewer 1]

  1. Comments:

 1) Unlike clinical diagnosis, which relies on symptoms and various tests, autopsy diagnosis does not depend on the direct visualization of infarcted myocardial tissue. 

-> After "Unlike clinical diagnosis", there should be a difference between autopsy diagnosis and clinical diagnosis. But, 'not depend on the direct visualization of infarcted myocardial tissue' is not a difference, because, also in clinical diagnosis, visualization of infarcted myocardial tissue is very rare.

(Reply) We have revised the sentences as follows in lines 45 to 52.

Clinical diagnosis facilitated by tools such as electrocardiograms, echocardiograms, coronary angiography, and cardiac biomarkers, which also allow for effective monitoring of disease status. Forensic diagnosis is challenging due to postmortem changes, making it impossible to use blood biomarkers commonly used in clinical settings. Therefore, the degree of coronary artery occlusion, the presence of thrombi, and ischemic changes or fibrosis of the myocardium are assessed during the autopsy. And, the changes of myocardium, the extent of fibrosis and inflammatory cell infiltration are examined through histopathological analysis [4,5].

2) Exosomes protect cargos such as proteins, DNA, and RNA from enzymatic degradation within their bilayer membrane and deliver them to target cells.

-> Is this sentence correct? Do you mean proteins, DNA, and RNA are cargos protected by exocomes? Or Exosomes are protection cargos?

(Reply) We have revised the sentences as follows in lines 61-3.

Exosomes protect biomolecules such as proteins, DNA and RNA from enzymatic degradation within their bilayer membrane and deliver these cargos to target cells.

  1. Material and methods

1) Please describe the criteria to diagnose the MI deaths. Although it was autopsy diagnosis by medical examiners, the description is required as inclusion criteria for this study.

Reply) We agree with the reviewer's opinion regarding the importance of accurate diagnosis. Accordingly, we have revised and supplemented the text in p.8, lines 224-9.

The MI group included autopsy cases of acute myocardial infarction based on autopsy findings, where significant coronary artery atherosclerosis and myocardial hypertrophy observed. Additionally, these MI cases were finally diagnosed when histopathological examination revealed either more than 70% coronary artery stenosis or the presence of a thrombus, along with marked interstitial fibrosis in the myocardium.

2) In the description about Table 1, Table 1 should contain information on sex, age, PMI, BMI, risk factors related to cardiovascular diseases, heart weight, and type of atherosclerotic lesion. But, PMI, risk factors related to cardiovascular disease, and heart weight are missing. Please add them to the Table 1.

Reply) We fully agree with the reviewer's appropriate suggestions. In accordance with the reviewer's comments, we have thoroughly revised Table 1.

3) AHA classification represents maturity and advance of coronary atherosclerosis. It is expected to be correlated with the degree of occlusion, but not the same. How about include the percent of occlusion in your description of coronary atherosclerosis? The criteria for this percent occlusion can be found in a Book by Buja and Butany, Cardiovascular pathology 5th edition, pp8-10.

4) If there was thrombus or visible infarct in the ventricular wall in any MI case, it can be a good positive control. Wasn't there?

Reply) In accordance with the reviewer's comments, we have updated the content of Table 1 to include the degree of stenosis. Additionally, we have marked with an asterisk (*) the two cases where a thrombus was identified in the Lesion type section (MI group cases 3 and 14). This information has also been added to the Table Legend

Reviewer 2 Report

Comments and Suggestions for Authors

It is one of the most important issues to correctly dignose the cause of  death. It is well known that postmortem dignsosis of eartly-onset ischemic heart disease in forensic autopsy cases because of the lack of specific morphological findings. Thus, there are lots of studies on potential biomarkers for the postmortem diagnosis of of ischemic heart disease in forensic autopsy cases. The authors examiend the exosomal microRNA in order to explore potential biomarkers for the postmortem diagnosi of ischemic heart disease in forensic autopsy cases.  Susequently, they found that exosomal microRNA 486-5p would have a ptential as one of the biomarkers for postmortem diagnosis of of ischemic heart disease. This study is quite interesting and valuable for the publication. However, there are several issues that should be resolved in the revision process.

#1. The authors should mention the influence of postmortem intervals on microRNA levels.

#2. The authors should cite the following two papers:

Sci Rep. 2024 Feb 19;14(1):4046. doi: 10.1038/s41598-024-54530-x.PMID: 38374168 

Sci Rep. 2021 Nov 8;11(1):21828. doi: 10.1038/s41598-021-01102-y.PMID: 34750390

Comments on the Quality of English Language

It is necessary of minor editing of English language

Author Response

[Reviewer 2]

It is one of the most important issues to correctly diagnose the cause of death. It is well known that postmortem diagnsosis of eartly-onset ischemic heart disease in forensic autopsy cases because of the lack of specific morphological findings. Thus, there are lots of studies on potential biomarkers for the postmortem diagnosis of of ischemic heart disease in forensic autopsy cases. The authors examiend the exosomal microRNA in order to explore potential biomarkers for the postmortem diagnosis of ischemic heart disease in forensic autopsy cases.  Susequently, they found that exosomal microRNA 486-5p would have a potential as one of the biomarkers for postmortem diagnosis of of ischemic heart disease. This study is quite interesting and valuable for the publication. However, there are several issues that should be resolved in the revision process.

#1. The authors should mention the influence of postmortem intervals on microRNA levels.

Reply) We greatly appreciate the reviewer's emphasis on the key points of our paper. In response, we have added the following content on p.2, lines 60-75:

In our previous study, we successfully isolated exosomes from postmortem blood (plasma) and pericardial fluid, finding that their morphology did not significantly differ from that of exosomes derived from living individuals. Additionally, Bioanalyzer Electropherogram Analysis confirmed that miRNAs of 20-200 nt in length were stably preserved in postmortem exosomes, even under degradation conditions. This suggests that exosomes may overcome the limitations of traditional biomarker candidates, which rapidly degrade after death."

#2. The authors should cite the following two papers:

Sci Rep. 2024 Feb 19;14(1):4046. doi: 10.1038/s41598-024-54530-x.PMID: 38374168 

Sci Rep. 2021 Nov 8;11(1):21828. doi: 10.1038/s41598-021-01102-y.PMID: 34750390

Reply) Both papers focus on identifying supplementary markers that can be used to diagnose acute ischemic heart disease during autopsy. We have incorporated the reviewer's suggestions into the Introduction sections, including appropriate references.